# Doppler Ultrasound Monitoring of Echogenicity in Asymptomatic Subcritical Carotid Stenosis and Assessment of Response to Oral Supplementation of Vitamin K2 (PLAK2 Randomized Controlled Trial)

**DOI:** 10.3390/diagnostics11020229

**Published:** 2021-02-03

**Authors:** Yamume Tshomba, Domenico Baccellieri, Niccolò Carta, Giuseppe Cilli, Vincenzo Ardita, Luca Apruzzi, Diletta Loschi, Andrea Kahlberg, Luca Bertoglio, Renata Castellano, Elisa Simonini, Felicita Andreotti, Roberto Chiesa

**Affiliations:** 1Department of Vascular Surgery, Fondazione Policlinico Universitario Gemelli IRCCS, Roma-Università Cattolica del Sacro Cuore, 00168 Rome, Italy; yamume.tshomba@unicatt.it; 2Department of Vascular Surgery, IRCCS Ospedale San Raffaele, Via Olgettina, 60, 20132 Milan, Italy; carta.niccolo@tiscali.it (N.C.); peppe.cilli@live.it (G.C.); ardita.vincenzo@hsr.it (V.A.); apruzzi.luca@hsr.it (L.A.); loschi.diletta@hsr.it (D.L.); kahlberg.andrea@hsr.it (A.K.); bertoglio.luca@hsr.it (L.B.); castellano.renata@hsr.it (R.C.); simonini.elisa@hsr.it (E.S.); chiesa.roberto@hsr.it (R.C.); 3Department of Cardiovascular Sciences, Fondazione Policlinico Universitario Gemelli IRCCS, Roma-Università Cattolica del Sacro Cuore, 00168 Rome, Italy; felicita.andreotti@unicatt.it

**Keywords:** atherosclerosis, carotid stenosis, doppler ultrasound vitamin K2, gray scale median (GSM), plaque composition

## Abstract

Background: Plaque composition may predict the evolution of carotid artery stenosis rather than its sole extent. The grey scale median (GSM) value is a reproducible and standardized value to report plaque echogenicity as an indirect measure of its composition. We monitored plaque composition in asymptomatic subcritical carotid stenosis and evaluated the effect of an oral modulating calcification factor (vitamin K2). Methods: Carotid plaque composition was assessed by GSM value. Monitoring the effects of standard therapy (acetylsalicylic acid and low–medium dosage statin) (acetylsalicylic acid (ASA) arm) or standard therapy plus vitamins K2 oral supplementation (ASA + K2 arm) over a 12 months period was conducted using an ultrasound scan in a prospective, open-label, randomized controlled trial (PLAK2). Results: Sixty patients on low–medium dosage statin therapy were enrolled and randomized (30 per arm) to either ASA + K2 or ASA alone. Thirty-seven patients (61.6%) showed at 12 months a stable plaque with a mean increase in the GSM value in respect to the baseline of 2.6% with no differences between the two study arms (*p* = 0.66). Fifteen patients (25%) showed an 8% GSM value reduction respect the baseline with no differences between the two study arms (*p* = 0.99). At multivariable analysis, the adjusted mean (95% confidence interval) GSM change per month from baseline was greater in the ASA + K2 arm (−0.55 points, *p* = 0.048) compared to ASA alone (−0.18 points, *p* = 0.529). Conclusions: Carotid plaque composition monitoring through GSM value represents a laborious procedure. Although its use may not be applied to everyday practice, a specific application consists in evaluating the effect of pharmacological therapy on plaque composition. This 12 months randomized trial showed that the majority of subcritical asymptomatic carotid plaque on treatment with low–medium dosage statin presented a stable or increased echogenicity. Although vitamin K2 beyond standard therapy did not determine a significant change in plaque composition, for those who presented with GSM reduction it did enhance a GSM monthly decline.

## 1. Introduction

Carotid artery stenosis is a major risk factor for stroke [1]. Current risk stratification to recommend surgical intervention beyond medical therapy is based on percent stenosis; this variable, however, does not completely reflect the real risk in many cases, as not all patients with significant stenosis will benefit from surgery, owing to stable plaques, whereas other patients without significant stenosis will experience neurological symptoms, owing to plaque vulnerability. Morphological characteristics, such as calcification or lipid content may be more important than stenosis itself to predict plaque instability [2,3]. Doppler ultrasound (US) assessment may provide additional features other than plaque stenosis and flow velocity. The gray scale median (GSM) represents a normalized and reproducible value of the plaque echogenicity thus reflecting its composition [4]. High values of GSM are associated with stable fibrotic or calcified plaque whereas low values are found in lipidic plaques more prone to become unstable [5,6,7]. Even if GSM value it is not widespread utilized, due to the laborious post-processing analysis, it has a potential role in monitoring plaque composition, in giving a reliable plaque risk stratification and in evaluating and tailoring pharmacological therapy. Although statin therapy in carotid stenosis leads to an overall increase in plaque echogenicity, interpreted as a stability factor [8], recent studies have shown a correlation between carotid calcification and cerebral ischemic lesions contributing to cognitive impairment and mortality [9,10,11,12]. Based on the plaque composition evaluation with doppler US and GSM application we monitored asymptomatic subcritical carotid artery stenosis on standard therapy and evaluated the effects of oral supplementation of vitamin K2 as a modulating factor in vascular calcification in 12 months trial [13,14,15,16,17,18,19].

## 2. Materials and Methods

### 2.1. Study Design

Plaque composition and evolution were assessed and monitored by ultrasound scan using the GSM value in a study cohort of asymptomatic subjects with subcritical carotid artery stenosis in a 12 months period. The further effect of oral supplementation of vitamin K2 as modulator factor in vascular calcification was assessed within a prospective, randomized, single-center, open-label, controlled trial. The study received institutional ethics committee approval (74/INT/2016) and is registered at clinicaltrials.gov (NCT02970084).

### 2.2. Patients

Adults with unilateral asymptomatic carotid atherosclerosis determining stenoses of 40% to 60% on treatment with low–medium dosage statin therapy (e.g., atorvastatin 20–40 mg qd, rosuvastatin 10–20 mg qd) were enrolled. Patients were subsequently randomized either to oral acetylsalicylic acid (ASA) 100 mg qd or to ASA 100 mg qd plus vitamin K2 800 mg qd (ASA + K2). Informed consent was obtained from each patient. The study protocol conforms to the ethical guidelines of the 1975 Declaration of Helsinki, as reflected in a priori approval by the institution’s human research committee.

### 2.3. Endpoint

Plaque evolution was evaluated with the observation of the GSM trend over time. We considered as the primary endpoint a significant evolution in GSM value intended as a change of 15 points at 12 months’ follow-up. We considered current literature [8,20] describing the effect of statin therapy on plaque echogenicity assessed by a GSM score that reported a plaque echogenicity increase from 16% to 29% at 12 months’ follow-up. We considered significant (*p* < 0.05) a reduction of 8% in the GSM value, scored as 15 points on a 0–190 scale, taking into account the natural trend toward increased plaque echogenicity (≥16%, i.e., ≥30 points) in our study population, given statin therapy. We assumed a significant (*p* < 0.05) GSM score reduction at 12 months of 33% among patients in the ASA + K2 arm versus 5% among those in the ASA arm, resulting in a calculated total of 60 patients required (alpha 0.05; beta 0.2, power 0.8).

### 2.4. Exclusion Criteria

Reduced life expectancy; age <18 or >80; enrollment in other clinical trial in the previous three months; hypersensitivity or known allergies to ASA, vitamin K2 and vitamin D; mastocytosis; history of salicylate-induced asthma; duodenal ulcer; hemorrhagic diathesis; severe liver failure; severe heart failure; concomitant treatment with methotrexate; pregnancy; hemolytic diseases; tumors; anticoagulant therapy; arteriovenous malformations; cerebral aneurysms; hypercalcemia; nephrolithiasis; chronic kidney disease (serum creatinine >1.2 mg/dL).

### 2.5. GSM Value

The GSM value intended as overall echogenicity was calculated from the frequency of bright, grey, and shadow pixels within the plaque. Its reliability lays on standardized measurements obtained by normalizing plaque structure to a linear scale set to two reference points such as blood and adventitia [21,22,23,24]. The analysis requires a strict protocol from doppler US setting to image acquisition and analysis. Plaque composition assessment through US doppler examination and GSM values analysis was conducted at enrollment, at six and at 12 months. To avoid operator-dependent variations, a single operator (NC), blinded to treatment allocation, performed all the examinations.

### 2.6. Doppler US Setting

It was standardized at 70 dB and 4.0 cm depth, with overall and time-gain compensation optimized for each patient, using an iU22 scanner (Philips North America Corporation, Andover, MA, USA) and an L12–5 (7.5 MHz) broadband linear transducer. A correct setup is shown in Figure 1 to reduce B-mode gain and to eliminate background noise, the blood should be completely black.

### 2.7. Image Acquisition

Complete B-mode axial and longitudinal images of the plaque are required to avoid over- or underestimate the overall plaque echogenicity in relation to plaque size. Particular attention should be taken on the longitudinal section in which both plaque and adventitia must be clearly visible. A complete morphologic and flow analysis examination should be obtained of the common carotid artery, carotid bulb and internal carotid artery respectively with a 90° and a 60° angle between US beam and longitudinal vessels axis [21]. Degree of stenosis was calculated based on morphological and velocimetric European Carotid Surgery Trial (ECST) criteria [25].

#### 2.7.1. Image Exportation and Analysis

Images were digitally exported into CD/DVD or USB support and analyzed using Adobe Photoshop CC 2015-16.0 software (Adobe, San Josè, CA, USA).

First step: color information was eliminated by transforming a red, green, blue (RGB), 8 bit image into a grey scale, 8 bit image (Figure 2).

Second step: adventitia was set as a reference point. After zooming on adventitia around 300%, an area of adventitia (at least 100 pixel) was selected with lasso tool. The median value was obtained via histogram (Window > Histogram) (Figure 3).

Third step: adventitia was normalized to a standard value of 190. Via curves (Image > Adjustments > Curves) the median value of selected adventitia was transformed into a standard value of 190 (Figure 4).

Fourth step: the entire image was normalized via curves with respect to the values previously obtained from normalization of adventitia. The plaque margins were outlined with the lasso tool and the median value showed on histogram was the GSM value (Figure 5).

In the case of non-homogeneous plaque, the GSM calculation should be focused on correctly set the adventitia as reference point in the second step and outline the entire plaque in the fourth step (Figure 6).

#### 2.7.2. Statistical Analyses

The primary endpoint was analyzed on the intent-to-treat exposed (ITT-E) population to avoid patient exclusion bias. This analysis included all patients undergoing randomization and receiving at least one day of study treatment. An additional “per protocol” (PP) analysis of the primary endpoint was done in all patients who completed treatment to evaluate the efficacy of treatment under optimal conditions. All secondary endpoint analyses were calculated on the ITT-E population. Results are expressed as medians (interquartile), means (95% confidence interval), or frequencies (%). Patient characteristics were compared by Mann–Whitney test or Chi-square/Fisher’s exact test, as appropriate. Mean changes in GSM over time (slopes) were estimated for each study arm and tested for significance. Comparisons of 12 months GSM reductions from baseline were stratified according to the following thresholds: 0 points, 15 points, or 7 points (median value). Univariate and multivariate linear regression models (fitted with random slope and intercept for each patient) were calculated to estimate factors associated with monthly and 12 months GSM mean change. All statistical tests were two-sided at a 5% level and performed using SAS Software (version 9.4, SAS Institute, Cary, NC, USA).

#### 2.7.3. Case Report Form (CRF)

Clinical data were collected in paper format and subsequently transferred to Excel files for statistical analyses. Data were rendered anonymous through codes based on progressive randomization number. Age, gender, height, weight, body mass index, smoking status, use of antidiabetic, lipid-lowering, and antihypertensive drugs were recorded. Agreement between the CRF recorded data and the source documents was checked by a monitoring team according to the International Council for Harmonisation/Good Clinical Practice guidelines.

## 3. Results

The study started in November 2016 and ended in May 2019. Sixty patients were enrolled and randomized (30 per arm) to either ASA + K2 or ASA alone (Figure 7). There were eight discontinuations: three in the ASA + K2 arm (for ischemic stroke, metrorrhagia, or loss to follow-up) and five in the ASA arm (for atrial fibrillation, gastro-esophagitis, lung cancer, lack of compliance, or loss to follow-up). The total study duration was 27 months.

We recorded seven adverse events, reported in Table 1: three among patients on ASA + K2 and four among among those on ASA alone.

Baseline patient’s clinical characteristics are reported in Table 2.

The quantitative assessment of plaque composition through GSM value showed that thirty-seven patients (61.6%) presented a stable echogenicity (intended as absence of 15 points GSM reduction) at 12 months with a mean increase in the GSM value in respect to the baseline of 2.1% (4.1 points) for patients on ASA + K2 versus 3.1% (5.9 points) for patients on ASA alone (*p* = 0.666). Fifteen patients (25%) presented a decrease in echogenicity (intended as 15 points in GSM value) at 12 months 8 (13.3%) with ASA + K2 versus 7 (11.7%) with ASA alone (*p* = 0.999) (Figure 8 and Figure 9).

The qualitative assessment of plaque composition showed that at baseline, hyperechoic plaques were found in 45% of the total population (27 patients): 13 in the ASA + K2 arm and 14 in the ASA arm (*p* = 0.999). At 12 months, no difference was observed in plaque-type distribution compared to baseline (*p* = 0.999), with hyperechoic plaques found in 48.3% of the total (29 patients): 14 in the ASA + K2 arm and 15 in the ASA arm (*p* = 0.999).

At multivariable analysis, the adjusted mean GSM change per month from baseline was greater in the ASA + K2 arm (−0.55 points (1.10, −0.01), *p* = 0.048 versus baseline) compared to the ASA arm (−0.18 points (−0.75, 0.39), *p* = 0.529 versus baseline) (Table 3).

Baseline GSM tended to be lower in the ASA + K2 arm than in the ASA arm (71.5 (64, 93) versus 89 (5, 113); *p* = 0.315). In the ASA + K2 versus ASA arm, GSM reduction tended to be greater at 6 months (−3.5 (−12, 4) versus −1.5 (−11, 8), *p* = 0.615) and 12 months (−5 (−15, 4) versus −1 (−13, 11), *p* = 0.641) (Figure 10). In the PP (per protocol) analysis, there were no statistical differences between arms with respect to GSM reduction.

Patients with a GSM reduction of 15 points, compared to those without, tended to be older (71 (53–76) versus 61 (55–68) years; *p* = 0.075) with a higher baseline GSM (94 (82–113) versus 68 (53–96); *p* = 0.013) (Table 4). Patients with a GSM reduction of 7 points, compared to those without, had a higher baseline GSM (89 (67.5–103.5) versus 67.5 (52–96.5); *p* = 0.034). At univariable and multivariable analysis, the 12 months mean change in GSM from baseline correlated significantly with higher baseline GSM (*p* = 0.012 and *p* = 0.011, respectively). Use of antihypertensive drugs was also significantly associated with a higher mean change in GSM from baseline (*p* = 0.036) (Table 5). Peripheral/coronary artery disease was significantly associated with a higher mean GSM change per month from baseline (*p* = 0.014) (Table 3).

## 4. Discussion

In monitoring subcritical carotid artery plaque, the main effort is made to detect precociously stenosis changes while barely considering its composition. The lipid and calcium contents are variably associated to the plaque vulnerability based on the concept of a linear relationship existing between echogenicity and plaque stability [26]. Several methods have been proposed to evaluate plaque composition from doppler US examination to more invasive computed tomography (CT) and magnetic resonance imaging (MRI), reserving histological examination in the case of surgery as a basis for comparison [8,27]. Doppler US examination avoids important CT and MRI limitations such as radiation exposure, claustrophobia, and high costs. Doppler US, in fact, is a fast, low-cost, and non-invasive examination which is optimal to identify, stratify, monitor, and assess the response to pharmacological therapy [28,29,30]. Some of the advantages are lost throughout the subsequent computer-assisted post-processing analysis. It is a laborious and time-consuming procedure that makes it unsuitable for daily use. Therefore, a widespread application may be considered useless unless it is applied in specific settings such as monitoring a pharmacological therapy. Among doppler US post-processing analysis, several techniques, such as intimal media thickness, integrated back scatter, grey scale median, coarseness (heterogeneity of the plaque), have been described; their comparability is debated as well as the reliability with respect to more invasive methods [8]. Moreover, the preferential use of a method depends on different elements: clinical habits, instrumentation and facilities access, cost, and time limits. The GSM value is a standardized and reproducible parameter, even if doppler US acquisition is affected by several limitations such as two-dimensional imaging, artifacts due to the heavily calcified plaque, ability of the operator, inter-observer variability, and arterial wall or probe oscillation during image acquisition [31]. The maximum effort should be focused on reducing all possible bias, through a rigid protocol both on image’s acquisition and on post-processing analysis.

A recent meta-analysis on the effect of statin therapy on the plaque composition showed a positive increase in the plaque echogenicity up to 35% in 12 months, even if the role of calcium content is not completely understood and still controversial [8]. Some authors state that it is a stabilizing factor, whereas others consider it a predictor of plaque vulnerability, as it has been recently associated to the presence of cerebral ischemic lesions and cognitive impairment [9,10]. Vitamin K2 (menaquinones) acts as cofactor in the post-translational modification of the gamma-glutamyl carboxylation of glutamate residues in matrix Gla-protein, enhancing its inhibitor role on vascular calcification. Based on recent studies that have showed the role of vitamin K2 oral supplementation in regulating the process of vascular calcification, the present randomized trial aims to evaluate its modulating effect on plaque composition beyond low–medium dosage statin therapy [13,14,15,16,17,18,19,32].

The present 12 months trial showed that the majority (61.6%) of subcritical carotid artery stenosis on statin therapy presented a stable or slightly increased echogenicity (considered as the absence of more than 8% or 15 points reduction in GSM value). In the entire population, calcium regression (considered as the presence of more than an 8% or 15 points reduction in the GSM value) at 12 months was observed in 25% of patients (15 patients): eight randomized to vitamin K2 supplementation and seven to ASA alone. Vitamin K2 oral supplementation, in addition to standard therapy with acetylsalicylic acid and statin, did not lead to a significant difference in calcium regression at 12 months; however, in those who presented with GSM reduction, it did enhance the GSM monthly decline compared to baseline (mean change −0.55 points per month, *p* = 0.048) which was not observed with standard therapy alone (mean change −0.18 points per month, *p* = 0.529). Patients with compared to those without a GSM reduction of 15 points tended to be older (*p* = 0.075) and had a higher baseline GSM (94 (82–113) versus 68 (53–96), *p* = 0.013), which may reflect an overexpressed calcification process. At multivariable analysis, patients with a 15 points GSM reduction had a higher prevalence of peripheral/coronary artery disease (*p* = 0.014) and more frequent use of antihypertensive drugs (*p* = 0.036), suggesting that arterial calcification is activated in patients with multiple atherosclerotic risk factors [33,34].

In the literature, the benefits of vitamin K2 integration are reported mostly in uremic patients [16,17,18,19], and in those with a dysregulated process, it leads to exceptional vascular calcification. This particular group of patients is characterized by a low intake of vitamin K2 and a high level of inactive matrix Gla-protein [35]. Thus, in atherosclerotic non-uremic patients with normal food intake, vitamin K2 supplementation may not correspond to a linear clinical effect on vascular calcification. The vulnerability of calcific plaques can be attributed to a compliance mismatch between the rigid mineral calcification and the artery wall tissue causing plaque rupture due to mechanical stress. However, the location of calcification in the vascular structure, whether intimal or medial, is important. Intimal calcification is mostly related to atherosclerosis and varies from small (5–10 μm) hydroxyapatite mineral crystals in early lesions to highly calcified plaques. Medial calcification occurs primarily in association with chronic kidney disease and diabetes, independently of atherosclerosis [36]. Thus, vitamin K2 supplementation in non-uremic atherosclerotic patients may not achieve the same benefit reached in uremic patients, reflecting the two distinct mechanisms of intimal and medial calcification. Moreover, in the literature the benefits of oral supplementation of vitamin K2 on the progression of atherosclerosis are not always clearly associated with calcification reduction [18,19]. Calcification is closely related to inflammation in atherosclerotic plaque. In vivo molecular imaging techniques, such as 18 F-sodium fluoride positron emission tomography (PET), may be helpful in revealing calcification process and thus active parietal inflammation. Due to the fact of its ability to precociously depict vulnerable plaque, considering that molecular calcifications anticipate macrocalcifications and clinical plaque rupture, it may have a role in stratify the clinical risk beyond the stenosis severity and plaque composition. Moreover, it may help to define and to monitor the possible anti-inflammatory effect of the modulating factors, such as vitamin K2, that are actively involved in parietal calcification process [37].

On the background of a linear relationship between plaque stability and echogenicity, we observed that a minority of patients treated with standard therapy showed a reduction in plaque echogenicity. This finding correlates with the ambiguous role of vascular calcification, that may represent an element of vulnerability in only a minority of patients [9,10,11,12,26,38]. In patients with abnormal upregulated calcification, a tailored targeted molecular therapy oriented toward the calcification process, such as vitamin K2 supplementation, may be effective.

In the ASA + K2 arm, two major adverse events were recorded, namely, ischemic stroke (not related to carotid plaque) and stable angina (chronic inducible myocardial ischemia) requiring percutaneous trans-luminal myocardial angioplasty, whereas in the ASA arm, one patient underwent carotid endarterectomy, due to the asymptomatic stenosis progression not associated with plaque composition changes, reflecting the study group’s high atherothrombotic burden.

Limitations of the present trial include the relatively small number of participants and the relatively short study duration. The GSM value, as expressed before, represents a considerable limitation as it may not be the most sensitive tool to evaluate calcification [27]. However, other variables, such as costs, invasiveness, center experience, and preference and patient adherence need to be taken into account when choosing other methods. The estimated effect size of vitamin K2 supplementation in an unselected asymptomatic population with subcritical stenoses may have been overoptimistic resulting in statistical underpower to meet the primary endpoint. Targeted study populations with known activated calcification process, such as patients with chronic kidney disease or with hyperechoic plaques, should be considered for future investigations.

## 5. Conclusions

Doppler US post-processing analysis of carotid plaque through GSM values may be helpful in monitoring plaque composition. The GSM value calculation is a demanding and time-consuming procedure; thus, its application in monitoring out-clinic patients may not be an accessible option. A tailored application should be considered in evaluating a pharmacological therapy. A real application in a 12 months randomized trial showed that the majority of subcritical asymptomatic carotid plaque, treated with low-medium dosage statin, presented either a stable or an increased echogenicity. Oral supplementation of vitamin K2 did not determine significant changes in plaque composition, but in those who presented with a GSM reduction, it did enhance GSM monthly decline. Vitamin K2 may have a specific role in patients with overexpressed calcification such as uremic patients.

## Figures and Tables

**Figure 1 diagnostics-11-00229-f001:**
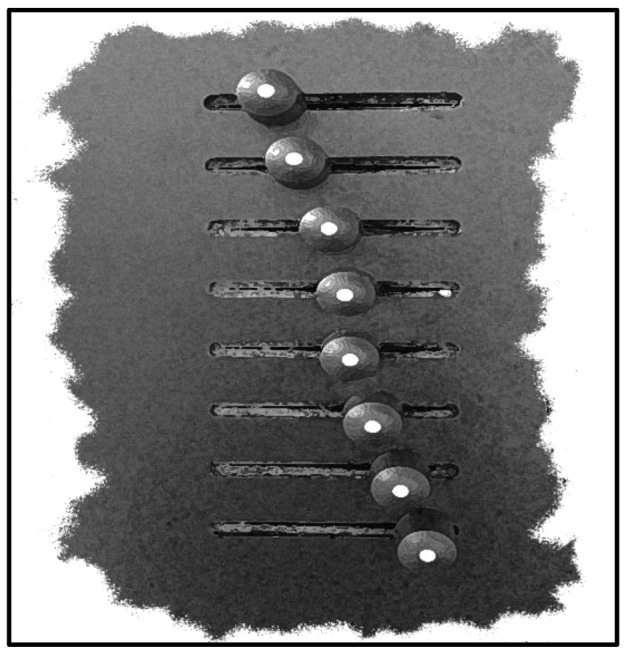
An example of correct doppler US set up with a gently curved line.

**Figure 2 diagnostics-11-00229-f002:**
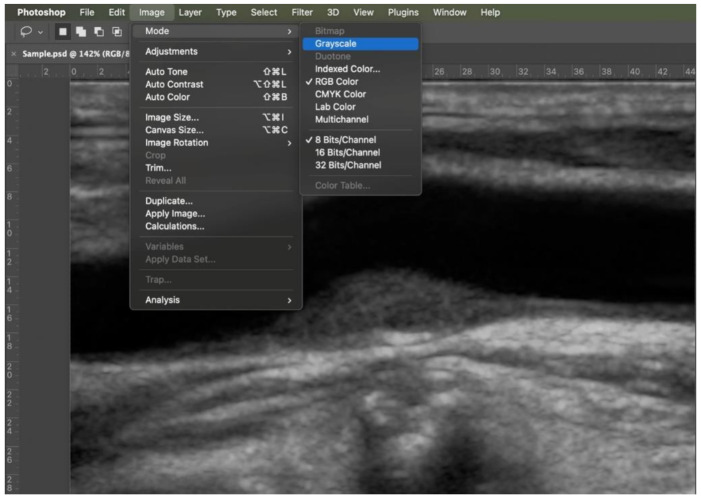
Step 1: transformation into a grey scale 8 bit image (Image > Mode).

**Figure 3 diagnostics-11-00229-f003:**
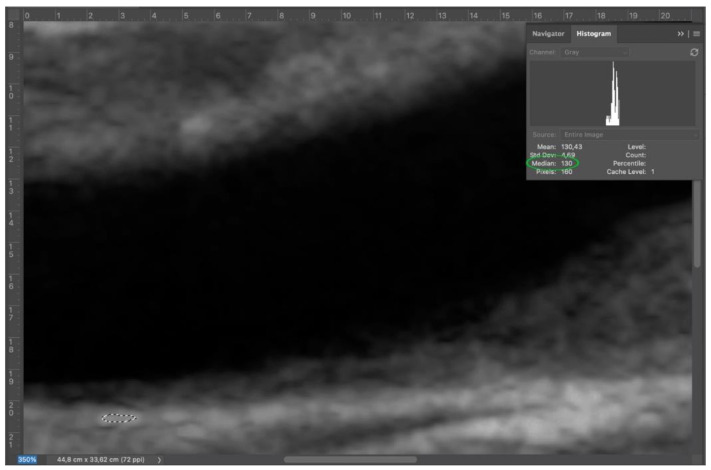
Step 2: select an adequate area of adventitia (dashed white circle), since it represents the key to perform a correct normalization.

**Figure 4 diagnostics-11-00229-f004:**
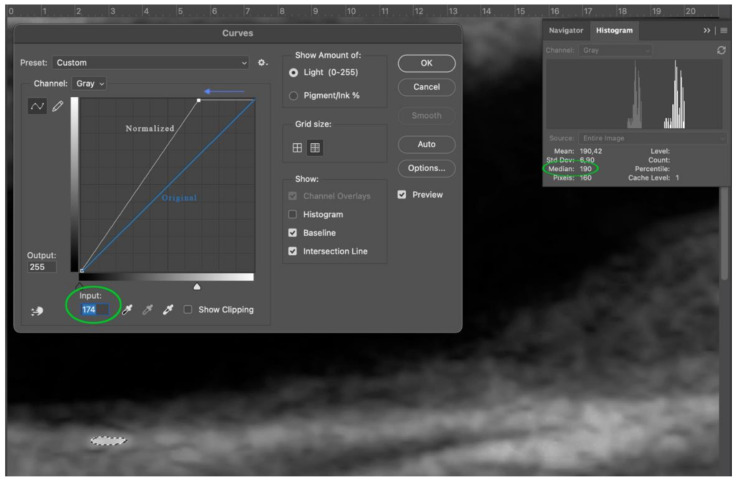
Step 3: move the apex of the straight line (blue) to the left in order to reach a median value of 190 on the histogram (green circle). During normalization only input value changes.

**Figure 5 diagnostics-11-00229-f005:**
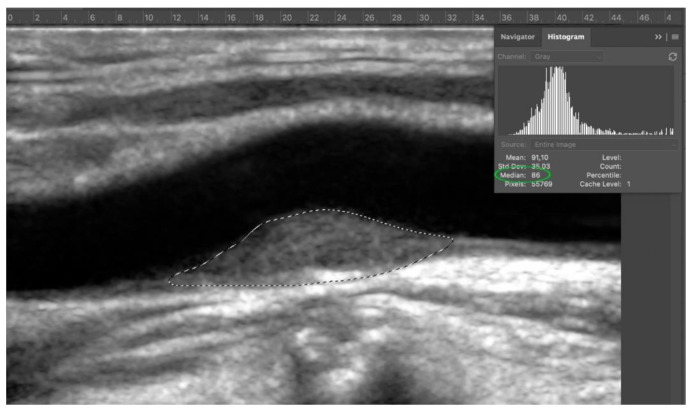
Step 4: grey scale median (GSM) of target plaque. In the example, the GSM is 86 (green circle).

**Figure 6 diagnostics-11-00229-f006:**
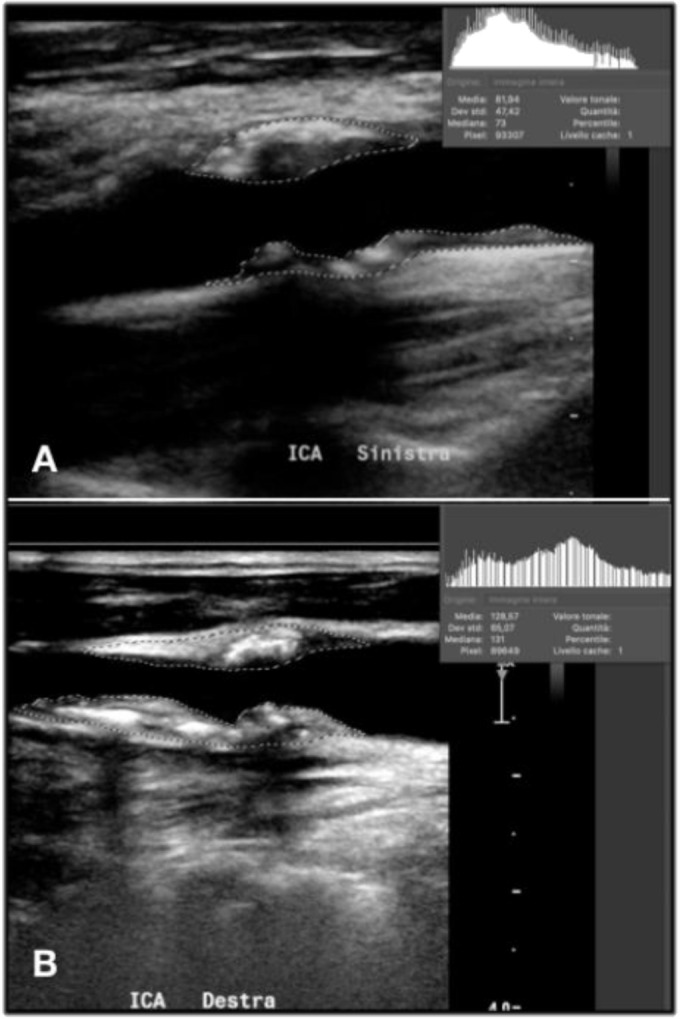
The GSM calculation in the case of an heterogenous composition (**A**) and an irregular profile (**B**).

**Figure 7 diagnostics-11-00229-f007:**
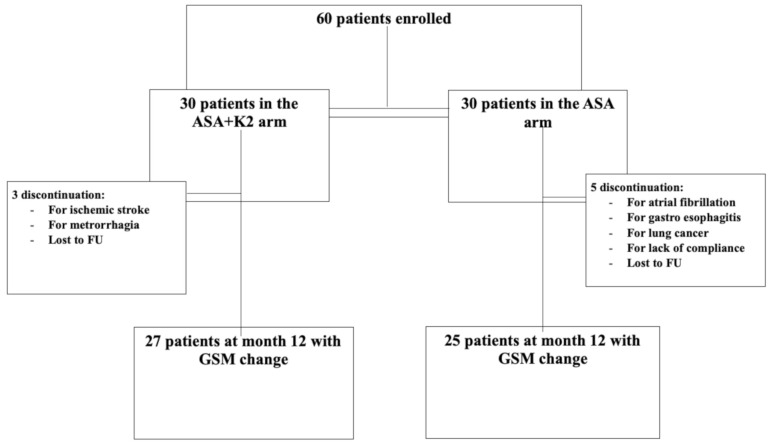
Study population.

**Figure 8 diagnostics-11-00229-f008:**
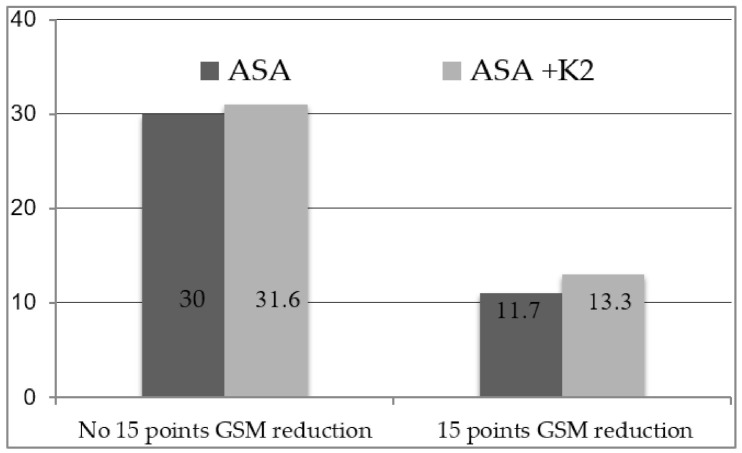
Proportion of subjects with and without a GSM reduction of 15 points at 12 months.

**Figure 9 diagnostics-11-00229-f009:**
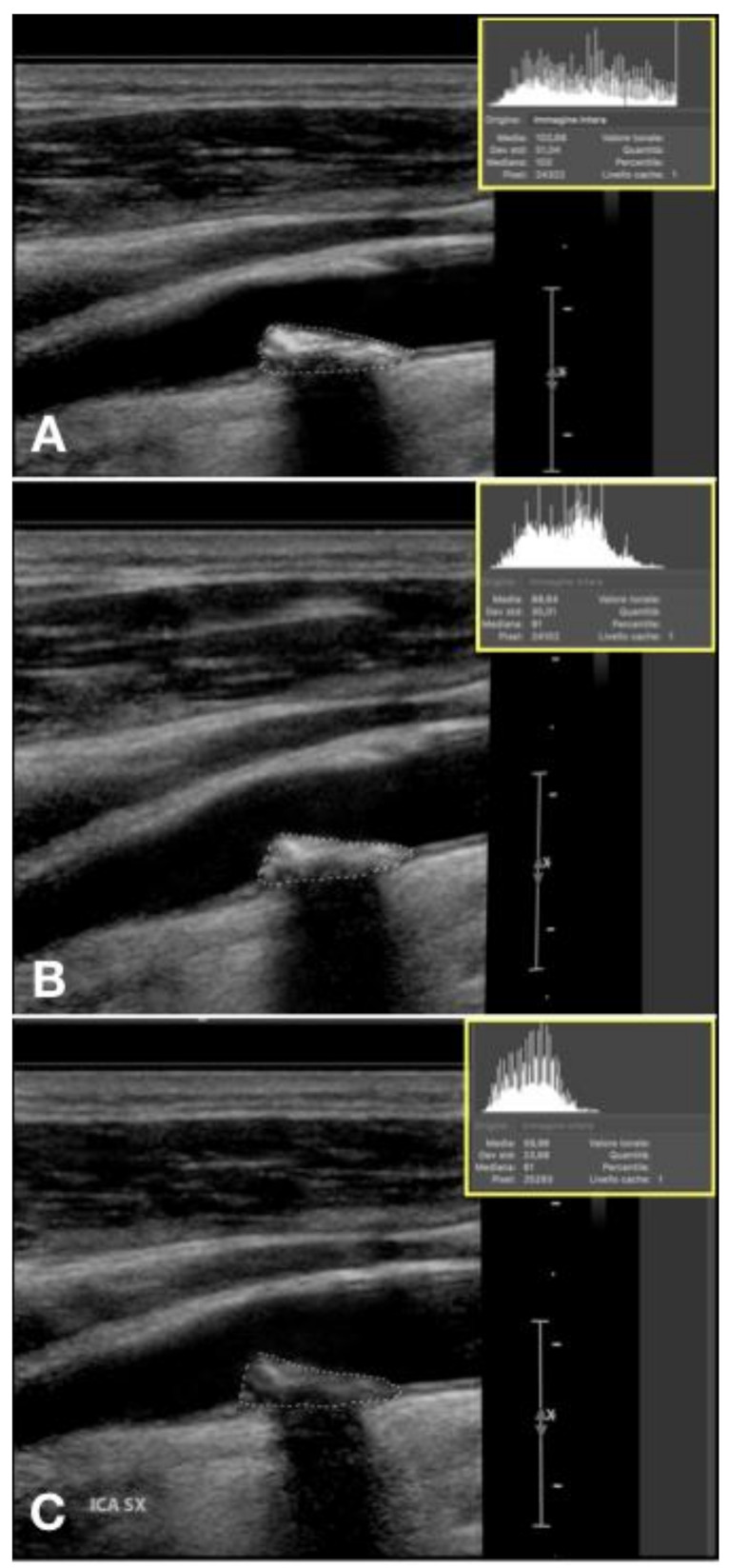
Plaque showing echogenicity decrease in 12 months treatment with vitamin K2. (**A**) start, GSM 103, (**B**) 6 months, GSM 91, (**C**) 12 months, GSM 61.

**Figure 10 diagnostics-11-00229-f010:**
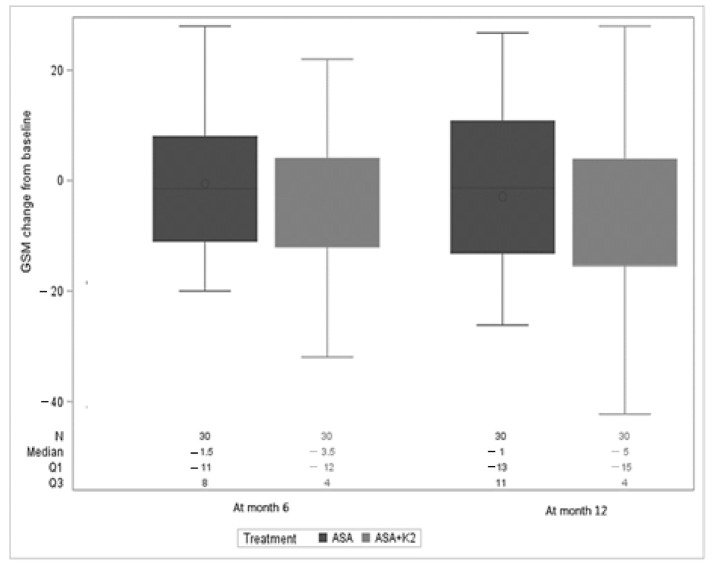
GSM change from baseline according to study arm.

**Table 1 diagnostics-11-00229-t001:** Adverse events recorded during the study.

Events	Arm	Time
Gastro-esophagitis	ASA	2 weeks
Atrial fibrillation	ASA	1 month
Ischemic stroke	ASA + K2	1 month
Carotid endarterectomy	ASA	2 months
Lung cancer	ASA	2 months
Metrorrhagia	ASA + K2	4 months
Percutaneous transluminal Coronary angioplasty	ASA + K2	9 months

**Table 2 diagnostics-11-00229-t002:** Baseline patient’s clinical characteristics.

Characteristics	Overall (*n* = 60)	ASA + K2 Arm (*n* = 30)	ASA Arm (*n* = 30)	*p*-Value
Age (years)	62.5 (55–71)	60 (53–68)	65 (57–73)	0.155
Gender				0.382
Female	16 (26.7%)	6 (20%)	10 (33.3%)	
Male	44 (73.3%)	24 (80%)	20 (66.7%)	
Weight (Kg)	74 (66–82.5)	74 (66–82)	76.5 (66–83)	0.739
Height (cm)	172 (166–178)	172 (167–176)	172 (165–180)	0.888
Body Mass Index (Kg/m^2^)	25.06 (23.24–27.44)	25.11 (23.1–27.4)	25.06 (23.44–28.34)	0.802
Smoke				0.751
Yes	15 (25%)	7 (23.4%)	8 (26.7%)	
Former	22 (36.7%)	13 (43.3%)	9 (30%)	
No	23 (38.3%)	10 (33.3%)	13 (43.3%)	
Arterial Hypertension				0.938
Yes	33 (55%)	16 (53.4%)	17 (56.7%)	
No	27 (45%)	14 (46.7%)	13 (43.3%)	
Dyslipidemia				0.999
No	29 (48.3%)	15 (50%)	14 (46.7%)	
Yes	31 (51.7%)	15 (50%)	16 (53.3%)	
Use of dyslipidemia drugs				0.389
Statin	54 (90%)	28 (93.4%)	26 (86.7%)	
Statin plus ezetimibe or omega3 fatty acid	6 (10%)	2 (6.6%)	4 (13.3)	
Diabetes				0.357
Yes	6 (1.7%)	2 (6.7%)	4 (13.3%)	
No	54 (91.5%)	28 (93.3%)	26 (86.7%)	
Peripheral/coronary artery disease				0.999
No	58 (96.7%)	29 (96.7%)	29 (96.7%)	
Yes	2 (3.3%)	1 (3.3%)	1 (3.3%)	
GSM	73.5 (62.5–100.5)	71.5 (64–93)	89 (55–113)	0.315
Stenosis (%)	40 (40–50)	40 (40–50)	40 (40–50)	0.628

**Table 3 diagnostics-11-00229-t003:** Univariable and multivariable mixed linear models performed to assess baseline factors associated with mean monthly GSM change from baseline (BL). Statistically significant *p*-value (<0.05) are highlighted.

Characteristics	Crude Mean Change per Month in GSM from BL (95%CI)	*p*-Value	Adjusted Mean Change per Month in GSM from BL (95%CI)	*p*-Value
ASA + K2 arm	−0.46 (−0.99;0.07) *p* = 0.096	0.570	−0.55 (−1.10; −0.01) ***p*** **= 0.048**	0.617
ASA arm	−0.26 (−0.82;0.30) *p* = 0.362		−0.18 (−0.75;0.39) *p* = 0.529	
Age (per year older)	0.03 (−0.08;0.14)	0.598	0.06 (−0.06;0.17)	0.356
BL GSM (per point higher)	0.01 (−0.03;0.05)	0.665	0.03 (−0.02;0.08)	0.184
Gender (Male versus Female)	−0.46 (−3.01;2.09)	0.716	−0.89 (−3.40;1.62)	0.480
Stenosis (per point increase)	0.03 (−0.11;0.16)	0.691	0.05 (−0.08;0.17)	0.460
Smoke (yes versus no)	0.25 (−2.03;2.54)	0.825	0.59 (−1.66;2.83)	0.602
BMI (per Kg/m^2^ increase)	0.05 (−0.29;0.38)	0.778	0.13 (−0.18;0.44)	0.403
Peripheral/coronary artery disease (yes versus no)	−4.89 (−10.66;0.88)	0.095	−7.2 (−12.91; −1.51)	**0.014**
BL plaque type (hyperechoic versus other)	0.31 (−1.92;2.53)	0.787	−0.55 (−2.88;1.79)	0.642

**Table 4 diagnostics-11-00229-t004:** Patients’ characteristics according to a GSM reduction of 15 points from baseline. Intent-to-treat (ITT) analysis (the significant *p*-value is unrelated to the treatment arm). Statistically significant *p*-value (<0.05) are highlighted.

Variable	GSM Reduction of 15 Points (*n* = 15)	No GSM Reduction of 15 Points (*n* = 45)	*p*-Value
Age (years)	71 (53–76)	61 (55–68)	0.075
Gender			0.999
Female	4 (26.7%)	12 (26.7%)	
Male	11 (73.3%)	33 (73.3%)	
Weight (Kg)	73 (66–81)	77 (67–83)	0.417
Height (cm)	170 (166–180)	172 (166–176)	0.791
Body Mass Index (Kg/m^2^)	24.19 (23.38–25.3)	25.4 (23.1–28.34)	0.140
GSM at baseline	94 (82–113)	68 (53–96)	**0.013**
Stenosis (%)	40 (40–50)	40 (40–50)	0.787

**Table 5 diagnostics-11-00229-t005:** Univariable and multivariable linear regression models performed to assess baseline factors associated with 12 months mean GSM change from baseline (BL). Statistically significant *p*-value (<0.05) are highlighted.

Characteristics	Crude 12 Month Mean Change in GSM from BL (95%CI)	*p*-Value	Adjusted 12 Month Mean Change in GSM from BL (95%CI)	*p*-Value
Arm (ASA + K2 versus ASA)	−2.77 (−11.20;5.66)	0.514	−7.56 (−16.63;1.51)	0.100
Age (per year older)	−0.23 (−0.65;0.20)	0.295	−0.36 (−0.85;0.13)	0.142
BL GSM (per point increase)	−0.19 (−0.34;−0.04)	**0.012**	−0.27 (−0.47;−0.06)	**0.011**
Gender (male versus female)	1.81 (7.75;11.36)	0.706	1.59 (−8.65;11.83)	0.756
Stenosis (per point increase)	−0.34 (−0.84;0.16)	0.174	−0.35 (−0.86;0.17)	0.183
Smoke (yes versus no)	−0.36 (−9.06;0.36)	0.934	1.01 (−8.58;10.60)	0.834
Body Mass Index (per Kg/m^2^ increase)	0.52 (−0.76;1.78)	0.422	−0.22 (−1.58;1.13)	0.741
Use of antidiabetic drugs (yes versus no)	2.67 (−12.61;17.96)	0.728	−9.80 (−27.51;7.91)	0.271
Use of antihypertensive drugs (yes versus no)	2.60 (−5.87;11.08)	0.541	10.61 (0.72;20.49)	**0.036**

## Data Availability

Y.T. this author takes responsibility for all aspects of the reliability and freedom from bias of the data presented and their discussed interpretation; D.B. this author takes responsibility for all aspects of the reliability and freedom from bias of the data presented and their discussed interpretation; N.C. this author takes responsibility for all aspects of the reliability and freedom from bias of the data presented and their discussed interpretation; C.G. this author takes responsibility for all aspects of the reliability and freedom from bias of the data presented and their discussed interpretation; V.A. this author takes responsibility for all aspects of the reliability and freedom from bias of the data presented and their discussed interpretation; L.A. this author takes responsibility for all aspects of the reliability and freedom from bias of the data presented and their discussed interpretation; D.L. this author takes responsibility for all aspects of the reliability and freedom from bias of the data presented and their discussed interpretation; A.K. this author takes responsibility for all aspects of the reliability and freedom from bias of the data presented and their discussed interpretation; L.B. this author takes responsibility for all aspects of the reliability and freedom from bias of the data presented and their discussed interpretation; R.C. this author takes responsibility for all aspects of the reliability and freedom from bias of the data presented and their discussed interpretation; E.S. this author takes responsibility for all aspects of the reliability and freedom from bias of the data presented and their discussed interpretation; F.A. this author takes responsibility for all aspects of the reliability and freedom from bias of the data presented and their discussed interpretation; R.C. this author takes responsibility for all aspects of the reliability and freedom from bias of the data presented and their discussed interpretation.

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
