# Peer review of "Doppler Ultrasound Monitoring of Echogenicity in Asymptomatic Subcritical Carotid Stenosis and Assessment of Response to Oral Supplementation of Vitamin K2 (PLAK2 Randomized Controlled Trial)"

_diagnostics, 2021, doi:10.3390/diagnostics11020229_

Round 1

Reviewer 1 Report

This study is a prospective randomized clinical trial.

The authors used dopper US and GSM values to monitor the therapeutic efficacy of standard therapy or standard therapy with vitamin K2 on patients with subcritical carotid artery stenosis.

They suggested that GSM method would be helpful in monitoring plaque composition and they found that low-medium dosage statin showed atheroprotective effect assessed by GSM method whereas vitamin K2 supplement showed no significant change.   

I have some comments about this article.

  1. Baseline clinical characteristics of enrolled patients in each treatment group (statin, statin +vit.K2) should be addressed as separate table
  2. Dose patients receive other atheroprotetice drugs such as omega-3 or fibrate in addition to standard therapy or vitamin K2? This point should be clarified to interpret the results.
  3. In discussion, as vitamin K2 is known to act on vascular calcification, what about the relation of GBM with specific vascular calcification image such as F-18 NaF PET?

Reviewer 2 Report

A well written manuscript focusing:

1) the role of US in the qualitative evaluation of carotid plaques.

2) the relative usefulness of VitK2 in the evolution of carotid plaques.

Two main criticisms:

1) check the format of all the references (the title of the journal before the title of the article)

2) just one plaque is shown in the 4 figures. The morphology of that plaque is perfect to show how to calculate its echogenicity (homogeneous, regular profile). in clinical practice, greater part of carotid plaques are not homogeneous and show irregular profile. Accordingly:

  • please,  include other(s) example(s) with not-homogenous plaques (at least one) to show how to calculate mean echogenicity 
  • please, include  pre- and post- treatment US images of at least one carotid plaque showing variations of echogenicity
